# Modelling and Simulation of Dissolution/Reprecipitation Technique for Low-Density Polyethene Using Solvent/Non-Solvent System

**Sharif H. Zein** [1,*], **Ali A. Hussain** [1], **Osman Y. Yansaneh** [1] and **A. A. Jalil** [2,3]

1 School of Engineering, Chemical Engineering, Faculty of Science and Engineering, University of Hull, Kingston Upon Hull HU6 7RX, UK
2 Center of Hydrogen Energy, Institute of Future Energy, Faculty of Engineering, Universiti Teknologi Malaysia, Johor Bahru 81310, Malaysia
3 Faculty of Chemical and Energy Engineering, Universiti Teknologi Malaysia, Johor Bahru 81310, Malaysia
* Correspondence: s.h.zein@hull.ac.uk; Tel.: +44-1482-466-753

**Abstract:** The global production and consumption of plastics have continued to increase. Plastics degrade slowly, causing persistent environmental pollution Developed waste plastic recycling methods are discussed in this report, with a focus on the dissolution/reprecipitation technique to restore low-density polyethene (LDPE) wastes. Aspen HYSYS is used to simulate the recycling of waste LDPE. Turpentine/petroleum ether (TURP/PetE) is chosen as solvent/non-solvent with fractions proved efficient through laboratory experiments. PetE is selected to be the non-solvent used for the precipitation of pure LDPE. The feedstock is assumed to be LDPE products containing additives such as dye. The simulation model developed estimated a pure LDPE precipitate recovery with a composition of 99% LDPE with a flowrate of 1024 tonnes per year. In addition, Aspen HYSYS could approximate a rough cost estimate that includes utility cost, installation cost and other factors. Technical challenges were eliminated, and several assumptions were taken into consideration to be able to simulate the process.

**Keywords:** biodiesel: LDPE waste; pollution; Aspen HYSYS; pure LDPE; cost analysis

## 1. Introduction

Plastic manufacturing has increased massively worldwide in recent decades [1]. Plastics are highly likely the most versatile material known to humankind. Since the first-scale production of synthetic polymers happened in the early 1940s [2], the manufacturing, consumption and rate of plastic waste have increased noticeably. Therefore, many researchers have focused on finding recycling routes in the past [3]. As such, an estimated 335 million tonnes of waste plastics were produced in 2016 globally, with 60 million tonnes generated in the European Union alone. It is estimated that an increased rate of 3.6% annually globally. It is estimated that only 26% of waste plastic is recycled, 36% is recovered with energy recovery processes such as incineration, and the remainder becomes disposed of in landfills. The incineration of such waste can release harmful gases, which can cause many environmental issues such as the formation of dioxins, fly ash, the production of nitrogen oxides and sulphur and other toxins [4]. In addition, plastic debris are widely noticed in the marine environment, but the exact quantity remains unknown. The issues surrounding these debris are becoming significant because of their persistence and effects on the ocean [5].

With plastics being a kind of polymers, synthetic plastics are usually designed to mimic the relevant properties of natural materials. Such polymers can be produced by synthesising primary chemicals from coal, oil, or natural gas [6]. Many other developments continue to increase the contribution of plastics as the primary material in the coming decades.

Plastics are originally derived from petroleum, and the reason for the increased usage of these materials can come from the ingenuity of chemical engineers. They recommend the manufacturing process. Their low cost also has its roots in the great quantity of the feedstock and the economic scale. For one barrel of petroleum, almost less than 5% is used for manufacturing polymers [7]. There are many uses of polymers, and these include solid moulded forms for automobile body parts, film packaging, TV cabinets, aircraft parts, coffee cups and foams, insulators, fibres for clothing and carpets, coating to change the appearance of other materials, adhesives, and many other uses [7]. The use of plastics is dominated mainly by packaging in Europe, followed in descending order for cosmetics and hygiene products, plastic bags, toys, 'other' and cutlery [8]. The properties of polymers are unique, and their origins come from the molecular composition of their significantly long chains in manufacturing products. Both processing, which is affected by orientation and flow and composition, including molecular size, chemical makeup, cross-linking and branching, are crucial to the estimated properties of the end product.

Most of the time, the waste streams are uncontaminated by other non-polymers or polymers. The product is disposed of at the end of life and turns into post-consumer waste [9]. Energy recovery is recommended if plastic waste is not recycled. Landfilling is the least preferred option due to its immense environmental effects [10].

The intense and increased production of polymers (plastics) is likely to lead to an increased rate of waste streams. However, these waste plastics, contaminated or, can be recycled, partially or 'fully' into new products [11]. Most waste plastic is disposed of through landfilling or incineration; as mentioned in their work, the amount recycled is considerably low. For that reason, process routes are investigated to treat the waste and produce petrochemical feedstock or fuels that can be very useful [12]. Chemical recovery involves recovered chemicals such as monomers that can be converted from plastic. This can occur through controlled thermal degradation or catalytic depolymerisation [3]. A sustainable and efficient treatment can occur with pyrolysis, producing a range of valuable hydrocarbons (HCs) that can be potentially used as energy or as a chemical feedstock [10,13]. Therefore, the dependency on non-renewable fossil fuels will be minimised while solving the landfilling problems [14].

Moreover, in contrast to metals and ceramics, recycling polymers is currently impossible without at least a few downgrading properties. However, that does not imply that nothing could improve the quality of products produced from recycled plastics up to the desired level [15]. Recycling plastic waste from many products, including appliances, textiles, automobile parts, films, and greenhouses, has been successful. The treatment and recycling of such wastes can be categorised into four major categories [16]. These include re-extrusion, mechanical, chemical and energy recovery. Each method offers a unique set of benefits that make these routes beneficial for specific applications, locations and/or requirements. Mechanical recycling involves the physical treatment, the direct reuse of uncontaminated plastics into a new product without losing their properties. The mechanical process usually requires shredding, crushing, grinding, or milling. This stage of the mechanical process technique is generally referred to as pre-treatment, as highlighted in work done by Yansaneh and Zein [10]. This will make the material more homogeneous and easier to blend with adding additives [15]. Chemical recycling and related treatment processes produce feedstock chemicals for the chemical industry. Energy recovery involves partial or complete oxidation of the material [3], where heat is produced with power, fuel (including oils), and chars besides by-products that are required to be disposed of, such as ash. This paper will investigate chemical recycling methods to revert waste plastics to its virgin state using modelling and simulation techniques. The simulation tool applied is Aspen HYSYS and the feedstock used is LDPE wastes. The results achieved pure LDPE (99% purity) with the aid of the reprecipitation technique, with Turpentine/petroleum ether (TURP/PetE) solvent implored, in line with their proven efficacy, which far outweighs the yield quantity obtained in other researched experimentations [17]. It confirms the effectiveness of the Turpentine/petroleum ether (TURP/PetE) solvent for pure polyethy-

lene production and the significant cost-effectiveness capability it is formulated with. The simulation of such and under the influence of PetE non-solvent is unprecedented, to the best of the researchers' knowledge, and this work is believed to add value in this field and related schools.

## 2. Method

### 2.1. Experimental Methods

As stated in Section 1, the techniques for separating different types of plastics rely on the differences in shape, colour, density, solubility, and physiochemical properties. Processes based on solubility include dissolving a series of incompatible polymers in a common solvent at various temperatures or in different solvents for one polymer to be separated each time [18]. Different methods are in place which is employed to recover the polymer after the dissolution step. It can be recovered either by adding a proper non-solvent (anti-solvent) or by rapid evaporation of the solvent. The recovery of the polymers using a non-solvent is the recommended route in this report.

The dissolution process has been applied successfully on a laboratory scale for recycling different kinds of plastics. Certain techniques are followed, and no influence on critical properties was noted for the technique followed with excellent molecular weight and mechanical property, retention characterised of the recycled polymers as studies show [19–21]. The steps followed include:

- Shredding the waste into smaller pieces and, if needed washing it with water prior to shredding or after as they may be.
- Initial separation of the preliminary mixture into two or more mixtures by floatation in a specific liquid or water.
- The addition of a solvent that explicitly dissolves only one of the polymers under certain conditions.
- Removal of the non-dissolved polymer through filtration. Addition of non-solvent to precipitate the dissolved polymer.
- Distillation of the solvent and non-solvent to separate them for reuse.
- Applying the same procedure for each polymer of the mixture.

Studies show that the initial selection of solvent and non-solvent systems suitable for the recovery of singly polymers is based on certain factors [19,21]. These factors include the necessary minimum ratio of solvent/non-solvent for precipitation, the dissolving ability of the solvent, and the sufficient separation of solvent/non-solvent mixture by distillation and the equivalent energy consumption, and the viscosity of the obtained polymer solutions. These preliminarily chosen solvents and non-solvents are found effective in the case of single polymers. LPPE, HDPE, and PP mixtures were experimentally analysed to investigate their suitability as per the selectivity of the solvent, depending on their capabilities [18]. As such, to dissolve only one polymer under certain conditions, the recovered polymers form, which needs to be suitable for feeding processing units, and the use of only on solvent/non-solvent system for all polymers becomes imperative. Since it was proved that the dissolution method is feasible and effective, researchers applied the techniques in a pilot unit.

The pilot unit operated for singly polymers as well as two-component polyolefin mixtures. Table 1 shows the chosen solvent and non-solvent and dissolution temperature for separating polyolefin mixtures [18].

**Table 1.** Selected solvent, non-solvent and dissolution temperature for the separation of polyolefin mixtures [18].

| Polymer | Solvent | Non-Solvent | S/Non-S Ratio | Temperature °C |
|---|---|---|---|---|
| **LDPE** | Xylene | Propanol-1 | 3:1 | 85 |
| **HDPE** | | | 3:1 | 100 |
| **PP** | | | 3:1 | 135 |

The pilot unit can treat up to 10 kg polymer mixture per batch and comprises two main vessels for dissolution, precipitation, and filter. The feed of solvents is pumped into the unit, and all units are insulated to avoid energy loss. A vacuum dryer is also in place to dry the recovered polymers and support facilities and materials such as heating oil, cooling, steam boiler, and nitrogen supply tank. The separation and reuse of the solvent and non-solvent occur in the installed packed distillation column, as emphasised by [18].

The same concept of this technique is considered in this paper, using a different solvent/non-solvent system called Dissolution/Reprecipitation Technique. The system chosen is simulated to investigate the possibility of the highest recoveries of products at large-scale rates and the effect of changing the temperatures on the recycled products' quality and recovery percentage. In addition, the simulation will allow a better understanding of the solvent/non-solvent ratio to the amount of waste fed into the system.

Solvent/Non-Solvent

Owing to recent research, using pure turpentine, turpentine/PetE and turpentine/benzene as a solvent with various fractions and PetE and n-hexane as non-solvents were investigated. The blend solvents were seen as an excellent solvent for all polyolefin, and the dissolution temperature was less than the pure solvent at the same time. Most recycled samples showed high reconditioning with no significant difference from the virgin materials. To perform the laboratory experiment, the physical properties of the pure and blend solvents were estimated, such as the Flory-Huggins interaction parameter and the solubility parameter in particular. This was to investigate the probability of the solvents dissolving the polymer. The Flory parameter was calculated using [17], presented in Equation (1):

$$X = X_s + \frac{V_1}{RT}(\delta_1 - \delta_2)^2 \tag{1}$$

$V_1$ is the liquid molar volume of the solvent, $\delta_1$ and $\delta_2$ are the solubility parameter of the solvent and the polymer, $R$ = gas constant, respectively, $T$ is temperature, and $X$ is a function of the temperature and mole fraction of the solvent and polymer. As for the solubility parameter ($\delta$), whereas for the blend solvents and PetE, the calculations are carried out using the following Equation (Equation (2) [22].

$$\delta = \left(\Delta H_v - \frac{RT}{V_m}\right)^{1/2} \tag{2}$$

The density of the solvents was measured using a simple glass pycnometer and was analysed by the gas pycnometer. $V_m$ = molar volume at cm$^3$/mol, $V_m = 1/\rho$, and $\rho$ = solvent density. The capillary viscometer method was used to determine the viscosities. The heat of vaporisation ($\Delta H_v$) is calculated using Equation (3) [23,24].

$$\Delta H_v = 0.026Tb^2 + 23.7Tb - 2950 \tag{3}$$

where $Tb$ is the boiling temperature in kelvin. Table 2 shows the estimated solubility and Flory-Huggins interaction parameters for the solvents and polymers, calculated by [17]. For the simulation and modelling purposes, this paper will only involve using LDPE wastes and turpentine/PetE as a solvent, and PetE as the non-solvent. LDPE is selected due to its

availability and regenerating wide uses [13]. In addition, the solvent/non-solvent results showed that this system achieved the highest recovery of pure LDPE in the laboratory environment.

**Table 2.** Solubility and Flory-Huggins interaction parameters for pure and mixed solvents [17].

| Materials | Solubility Parameter | $X_{12}$ | Mixed Solvent Solubility Parameter |
|:---:|:---:|:---:|:---:|
| **Turpentine** | 8.08 | 0.35 | (0.25 + 0.75) |
| **PetE(A)** | 6.32 | 1.03 | 6.76 |
| **PetE(B)** | 6.79 | 0.71 | 7.11 |
| **PetE(C)** | 7.67 | 0.38 | 7.77 |

*2.2. Modelling and Simulation*

Using Aspen HYSYS to simulate the dissolution of waste plastic can have many advantages. Aspen HYSYS contains huge components libraries, models for various processing units and equipment, equations of states and other required parameters that are crucial in a large-scale process. This is supported by work carried out by [25]. Keep in mind that Aspen HYSYS cannot be more accurate than the systems and models behind it. Implementing models in Aspen HYSYS can be very useful in preliminary studies such as vessel sizing, energy consumption, and cost estimation. While developing the model, certain assumptions were taken into consideration, and as indicated in related studies of different sources, including [10,13,26]:

- Steady-state process.
- Additives present in LDPE (0.1%).
- Tray column is used in a distillation column with 100% tray efficiency. Small traces of plasticisers, flame retardants, antioxidants and thermal stabilisers components in streams are neglected.
- Waste LDPE fed into the system is shredded into tiny pieces.
- The solvent is recycled back to the dissolution tank.
- Blended stream is both liquid and vapour phase.

There are many methods for implementing experimental methods into Aspen HYSYS, specifically regarding compositions. However, the Aspen HYSYS databank does not contain any polymer. Hence, LDPE, dye, turpentine + PetE and PetE are added to the simulation as hypothetical components. For Aspen HYSYS to estimate the physical and chemical properties of LDPE, the stipulation of three properties is required: molecular weight, density, and normal boiling point [27]. These properties were added to the hypothetical component manager. They then became the basis on which the software predicted all the necessary information the polymer used to simulate the process accurately [28].

The Peng-Robinson fluid package was used in the simulation since it fits the material specifications used in this process. This gives rise to the fact that the Peng-Robinson model is perfect for vapour-liquid equilibrium calculations and liquid densities for HC systems which is the most suitable for the dissolution/reprecipitation of LDPE [25]. In this paper, multiple enhancements to the original Peng-Robinson model were created to extend its applicability range and improve its estimations for some non-ideal systems. The Peng-Robinson property package is rigorously capable of solving any single, two, or three-phase system with the highest degree of reliability and efficiency. It applies to a wide range of conditions. For example, ref. [29] researched a similar case wherein Peng-Robinson thermodynamic package is immensely vital for oil and gas processes and related petrochemical procedures, certified by Property Method Selection (APMS).

In the modelling and simulation of the dissolution of LDPE waste-recovered-various-products, the Aspen HYSYS model developed in this paper was based on the dissolution/reprecipitation technique for waste polyolefin recycling using a new, pure, and blended organic solvent report [17]. The solvent and non-solvent (Turpentine/PetE) se-

lected in the simulation process were chosen owing to their efficiency in yields [17,30]. Hence, a 99% purity of pure LDPE was obtained. They showed the highest recovery percentage in yields compared to the other two solvents in the experiment, as depicted in Table 2 and emphasised in work carried out by [17]. The mixed solvent Turpentine/PetE in a ratio of 0.5:0.5 were considered suitable for all types of waste polymers used in the experiment [17]. As such, the full dissolution of LDPE in the solvent occurred at a temperature of 80 °C. For the simulation, the dissolution temperature was set at 120 °C to ensure full dissolution without risking the polymer chains breaking due to excessive heat applications. The feed of waste plastic is assumed to be only LDPE consisting of a fraction of additives such as dye, as highlighted in this paper among the study's assumptions, that will be heated up to 120 °C before blending it with the solvent. The additives' properties were firmly estimated through Aspen HYSYS but supported by literature on the aspect of such feedstock having additives and related impurities [10,13,26]. As such, Table 3 shows the properties of the feedstock and the properties of the mixed solvent used in the simulation model.

**Table 3.** Properties of feedstock and mixed solvents.

| Solvent | Molecular Weight | Boiling Point (°C) | Density (kg/m³) |
|---|---|---|---|
| LDPE | 28.05 | 106 | 930 |
| Additives (dye) | 307.4 | 350 | 865.2 |
| Turpentine + PetE | 117 | 120–135 | 810 |

The non-solvent is also heated before being blended with LDPE before entering a tank where the dissolution is assumed to take place. A stirrer is assumed to be placed in the tank to ensure full dissolution. The blended feed consists of both liquid and vapour phases. LDPE and traces of the solvent blend are evaporated and sent into the first distillation column, while most of the solvent and traces of LDPE are recycled back to be mixed with the original feed. It is noted that the dissolution process dissolved the fraction of additives in LDPE before further processing. Vaporised LDPE is then cooled to a temperature of 80 °C to change its phase into a liquid phase for an easier distillation process. Table 4 shows the compositions of the liquid phase feed exiting the cooler after the dissolution in a solvent.

**Table 4.** Compositions of LDPE, Dye and solvent.

| Component | Mole Fractions |
|---|---|
| LDPE | 0.5525 |
| Dye | 0 |
| Turpentine + PetE | 0.4475 |

The purpose of distillation in this process is to filter LDPE from the remaining solvent in the cooled stream. The full reflux condenser concentrated the LDPE, which is the vapour phase exiting at a temperature of 105 °C and then cooled to a temperature of 75 °C to prepare it for blending with the non-solvent to achieve reprecipitation.

The non-solvent selected for this process was PetE since it was observed that it is a very good precipitator for the polymers used in the laboratory environment [17]. The non-solvent is heated to a temperature of 90 °C to blend with the filtrate from the first distillation column. Experiments have shown that the dissolution temperature of LDPE in PetE takes place at a temperature of 70 °C with a recovery percentage of 98% [17]. It is assumed that no second vessel is required, and the blended stream of LDPE and non-solvent are sent to the second distillation column where the reprecipitation will occur. The precipitated pure LDPE exits the bottom of the column, where the non-solvent and all other traces of components are evaporated and condensed with a full reflux condenser.

## 3. Results and Discussion

The Aspen HYSYS model in this study for the dissolution and reprecipitation of waste LDPE is referenced to [17] work to simulate a process that converts contaminated LDPE to its pure form for reuse or converting it into fuels. The results obtained from the simulation of the dissolution and reprecipitation of waste LDPE were compared with experimental data to evaluate the performance of the developed model. The developed model involves the simulation of a considerably large-scale process compared to the experimental data to understand the performance of such a process on a large scale since it is needed to prevent further harm to the environment. Table 5 shows the flow rate of each component in the filtration column.

**Table 5.** Feed properties for the filtration process.

| Component | Flow Rate (kg/h) |
|---|---|
| LDPE | 116.3 |
| Dye | 0.0011 |
| Turpentine + PetE | 392.8 |
| Total | 509 |

The non-solvent and additives exit the column as vapour from the total reflux condenser. Table 6 shows the flow rate and composition of the feed stream entering the precipitation column.

**Table 6.** Feed stream components flow rate.

| Component | Flow Rate (kg/h) |
|---|---|
| LDPE | 116 |
| Additives (dye) | 0 |
| Turpentine + PetE | 0.979 |
| PetE | 300 |
| Total | 417 |

The products obtained are shown in Table 7. The results show that the dissolution process is considered very good. The simulation model showed better recovery of LDPE than the results obtained from experimental methods [17]. Filtrate exited the reflux condenser at 105 °C as vapour at atmospheric pressure. The composition of the filtrate stream consists of 99.7% LDPE and only a negligible amount of 0.2% of the solvent (Turpentine/PetE) used. Note that the residues of PetE have the potential for further recycling and recovery processes for reuse, which is vital at an industrial scale [31].

**Table 7.** Mass recovery rates obtained from the simulation model.

| Component | Filtrate | Solvent + Additives |
|---|---|---|
| Flow rate (kg/h) | 117 | 0.179 |
| Recovery% | | |
| LDPE | 99.8 | 0.17 |
| Additives (dye) | 1.16 | 100 |
| Turpentine + PetE | 0.25 | 99.7 |

LDPE filtrate is then cooled to a temperature of 75 °C to reach its liquid phase before blending it with the non-solvent. Larrow and Jacobsen [32] applied the same cooling

temperature to cool down a similar homogeneous mixture to 75 °C over 60 min or less and were kept at such temperature (±5 °C) for 120 min. PetE, which is selected to act as a non-solvent in the reprecipitation pure liquid LDPE process, is added to the process with an LDPE/non-solvent ratio of 1:3. PetE is heated to a temperature of 89 °C higher than the dissolution temperature found in the literature. In addition, the temperatures set are required to be below the melting point of LDPE to avoid any degradation in polymer chains. LDPE filtrate stream and the heated non-solvent stream are then blended and sent to the reprecipitation column, where pure LDPE is recovered from the bottom of the column as a precipitate at a temperature of 105 °C.

Table 8 shows the composition of the products recovered as a precipitate from the column. 112 kg/h of pure LDPE is recovered with a higher recovery rate compared to the experimental data. The high purity of the recovered LDPE shows that changing specifications of the process can lead to higher recovery rates with keeping in mind the chance of polymer degradation.

**Table 8.** Precipitated product stream composition and recovery.

| Product | Mole Fractions | Recovery% in Precipitate Stream |
| --- | --- | --- |
| LDPE | 0.997 | 96% |
| Additives (dye) | 0 | |
| Turpentine + PetE | 0.0021 | |
| PetE | 0.0005 | 0.07% |

The simulation model proved that it is feasible to treat large amounts of waste plastics using a solvent/non-solvent to recover pure polymer precipitate that can be used to generate fuel and energy. The process can produce 117 kg/h of pure LDPE, meaning it can produce not less than 1020 tonnes of pure LDPE per year if productions are non-stop throughout. Such a production scale reveals the beneficial contribution to environmental conditions as more non-liquid products are being yielded. Table 9 shows the conditions of the precipitated pure LDPE recovered from the reprecipitating distillation column. Figure 1 shows the process flow diagram of the process simulated.

**Table 9.** Conditions of pure precipitated LDPE.

| Stream Name | Pure Precipitated LDPE | Liquid Phase |
| --- | --- | --- |
| Vapour/Phase Fraction | 0.000 | 1.000 |
| Temperature (°C) | 105.4 | 105.4 |
| Pressure (kPa) | 100.0 | 100.0 |
| Molar Flow (kgmole/h) | 3.986 | 3.986 |
| Mass Flow (kg/h) | 112.7 | 112.7 |
| Std Ideal Liq Vol Flow (m$^3$/h) | 0.1214 | 0.1214 |
| Molar Enthalpy (kJ/kgmole) | $-8.268 \times 10^4$ | $-8.268 \times 10^4$ |
| Molar Entropy (kJ/kgmole-C) | $-164.2$ | $-164.2$ |
| Heat Flow (kJ/h) | $-3.295 \times 10^5$ | $-3.295 \times 10^5$ |
| Liquid Vol Flow @Std Cond (m$^3$/h) | 0.1211 | 0.1211 |
| Fluid Package | Basis $-1$ | |

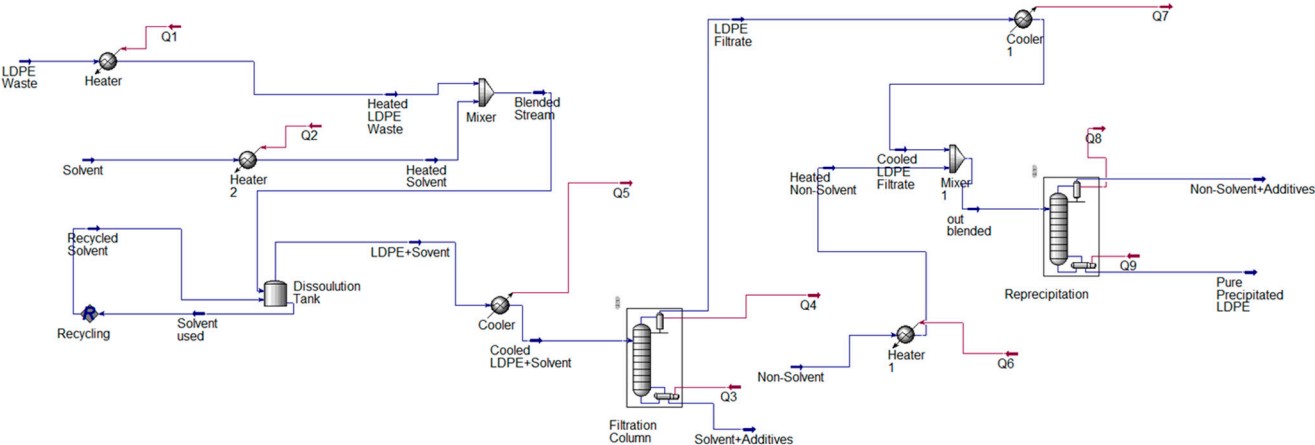

**Figure 1.** Dissolution/reprecipitation process flow diagram with precipitate collected at 105.4 °C.

The physical properties of the precipitated LDPE, such as boiling point and melting point, can provide crucial information that can help identify the product to establish its purity. Numerous normalised distillation tests determine the boiling point distribution of fuels [33]. Figure 2 presents some of the most common standard test methods used in the distillation of products. True Boiling Point (TBP) distillation is one of the most common investigational techniques for determining oil properties. ASTM D86-96, which is executed under atmospheric pressure and is used to determine the boiling point distribution of light petroleum fractions. ASTM D1160 is for heavy petroleum fractions while ASTM D2887 is to determine the TBP of fractions other than gasoline.

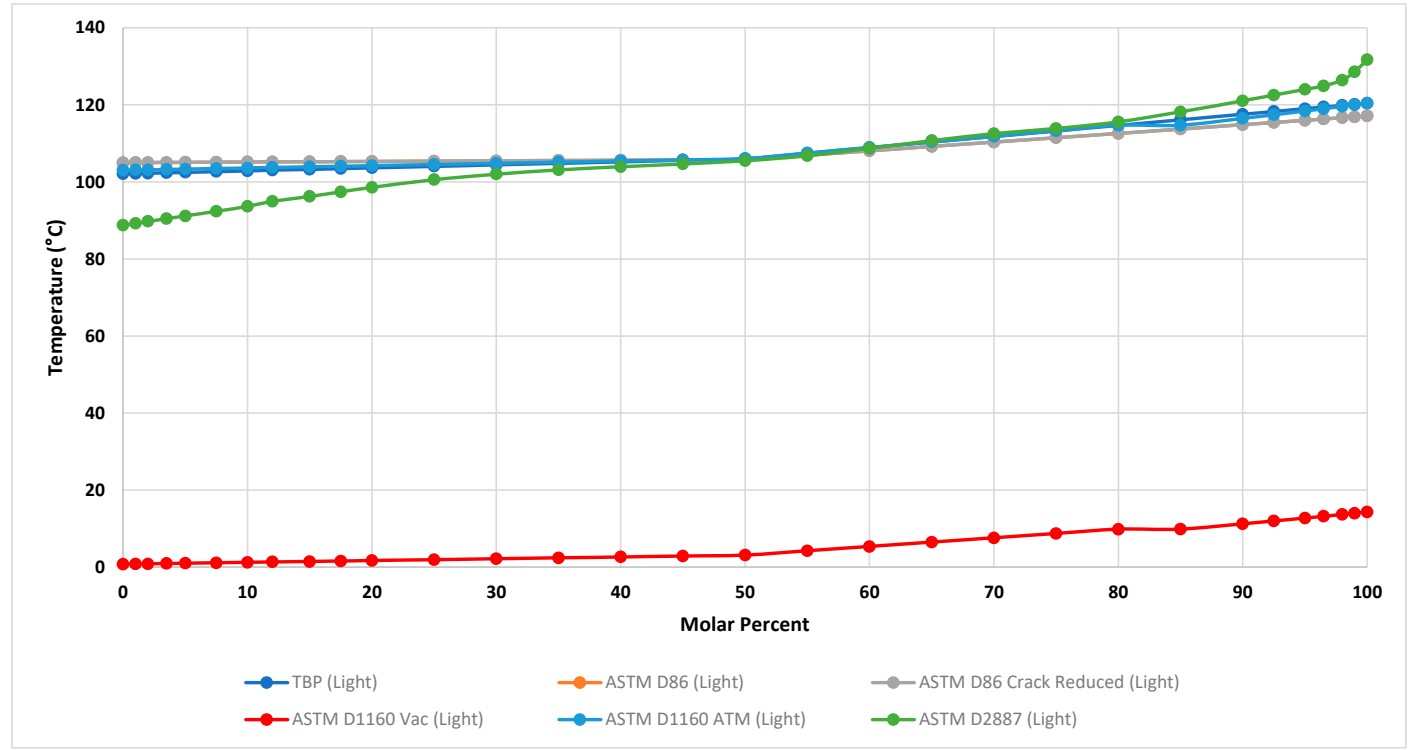

**Figure 2.** Boiling point curves of precipitated LDPE.

When the molecular size of a macromolecule or polymer is very large, the compound usually melts at a high temperature before the boiling point is reached. This is supported in work carried out by The Open University [34], where they emphasised the aspect of high-performance polymers- more resistant to high temperatures with a firm moduli

nomenclature. This can be seen from the plots as the mole percentage of the product increases, the melting point temperature gradually reaches 120 °C. The plot shows the ASTM distillation values, which refer to a diverse group of different ASTM international standards, supported by work carried out by Ferris & Rothamer [35]. The standards use distillation volatility characteristics of the substrate to determine the adherence of the substrate to the standard. ASTM D86 is a test method for the distillation of petroleum products at atmospheric pressure, which can be useful in the case of the products obtained since the precipitated LDPE can be turned into fuel [35]. As such, the D1160 test method is also plotted for the distillation of petroleum products at reduced pressure, whether it is vacuum or only reduced. The ASTM D1160 test method addresses the establishment, at minimised pressures, of the relevant range of boiling points of the said petroleum products [36]. The plot also shows that the process is difficult under vacuum distillation conditions since the boiling temperatures drop to 18 °C, which increases the complexity of the separation process.

Figure 3 shows the molecular weight plot for the pure LDPE precipitated stream. The plot shows that as the mole percentage of the stream increases, the molecular weight decreases gradually, supported in research work carried out by Modi et al. [37]. The feed pressure and composition within the system are believed to be responsible for this [38]. As the mole per cent reaches 50%, the molecular weight is 30 kg/mol. A steady increase is noticed in the molecular weight as the mole percentage increases after 50%. The highest molecular weight reached is approximately 120 kg/mol.

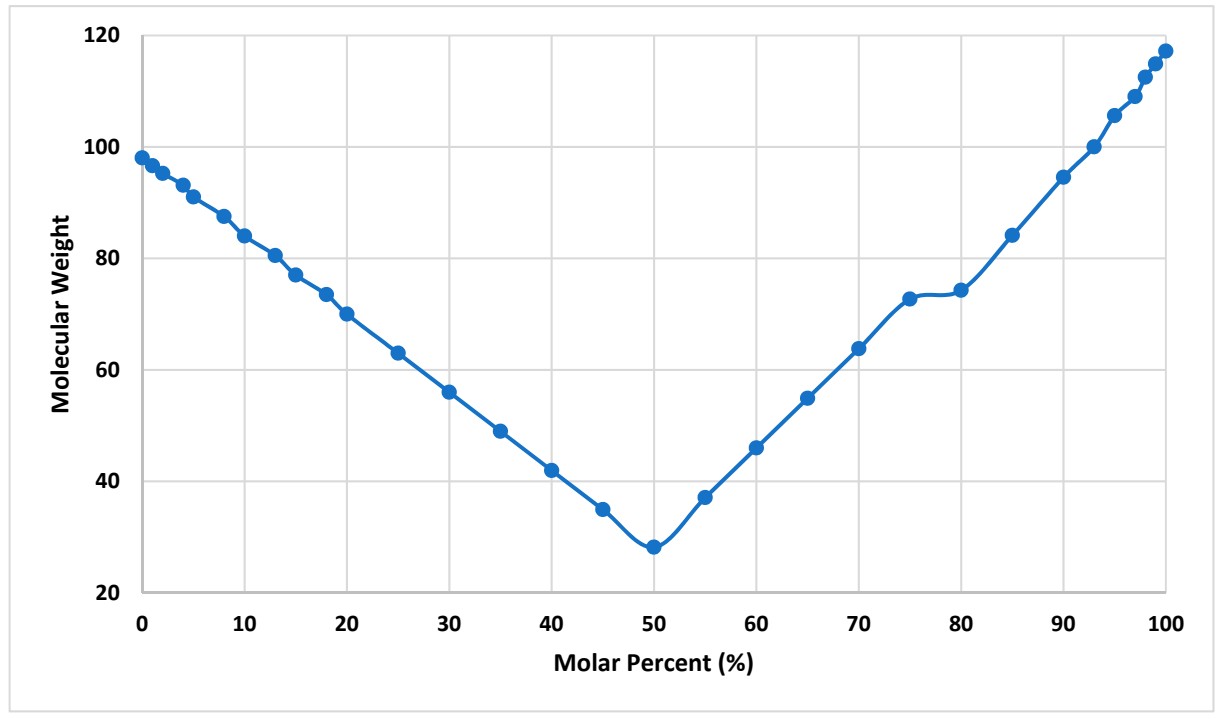

**Figure 3.** Molecular weight of precipitated pure LDPE plot.

The increase in molecular weight from its original weight can imply that the product's strength increased, which is evidence that the product of the recycled LDPE has better strength. Research has shown that high molecular weight means strength and toughness, and chemical stress crack resistance is increasing [39]. This is down to the fact that impurities have been filtered via the filtration column to a large extent as the generated steam condenses into LDPE, as depicted in Table 10. As such, the vast difference of (116.537–28.230) mole weight between the Reboiler and Condenser gives an indication of the filtered impurities and the nominal size left or small traces shared in the assumptions made in this study earlier.

**Table 10.** Column properties vs. tray position from top for steam.

| Component | Surface Ten [dyne/cm] | Mole Weight (Vap) | Density (Vap) [kgmole/m³] | Viscosity (Vap) [cP] | Therm Cond (Vap) [W/m-K] | Heat Cap (Vap) [kJ/kgmole-C] |
|---|---|---|---|---|---|---|
| Condenser | 21.451 | 28.230 | 0.923 | 0.005 | 0.010 | 1.167 |
| Tray 1 | 21.436 | 28.363 | 0.928 | 0.005 | 0.010 | 1.171 |
| Tray 2 | 21.409 | 28.597 | 0.935 | 0.005 | 0.010 | 1.176 |
| Tray 3 | 21.362 | 29.007 | 0.948 | 0.005 | 0.010 | 1.187 |
| Tray 4 | 21.279 | 29.729 | 0.971 | 0.005 | 0.010 | 1.204 |
| Tray 5 | 21.130 | 31.010 | 1.012 | 0.005 | 0.010 | 1.233 |
| Tray 6 | 20.859 | 33.305 | 1.085 | 0.005 | 0.010 | 1.280 |
| Tray 7 | 20.365 | 37.460 | 1.217 | 0.005 | 0.011 | 1.351 |
| Tray 8 | 19.512 | 44.947 | 1.451 | 0.006 | 0.012 | 1.449 |
| Tray 9 | 18.272 | 57.575 | 1.838 | 0.006 | 0.013 | 1.564 |
| Tray 10 | 16.950 | 75.035 | 2.364 | 0.007 | 0.015 | 1.668 |
| Tray 11 | 15.949 | 92.368 | 2.877 | 0.007 | 0.016 | 1.739 |
| Tray 12 | 15.370 | 104.569 | 3.234 | 0.008 | 0.017 | 1.778 |
| Tray 13 | 15.085 | 111.273 | 3.429 | 0.008 | 0.018 | 1.797 |
| Tray 14 | 14.956 | 114.480 | 3.523 | 0.008 | 0.018 | 1.805 |
| Tray 15 | 14.899 | 115.914 | 3.564 | 0.008 | 0.018 | 1.809 |
| Reboiler | 14.875 | 116.537 | 3.582 | 0.008 | 0.018 | 1.810 |

The pivotal role in eliminating the impurities can also be seen with the density (Vap) component with 3.582 kgmole/m³ at the Reboiler and less than 1.0 kgmole/m³ at the condensing column.

The critical temperature plot of the product stream was also generated by Aspen HYSYS to understand the products' limits, as seen in Figure 4. The critical temperature of a substance refers to the temperature at and above which steam of the affected material cannot be made liquid, irrespective of the amount of pressure incurred [40]. The plot shows the mass % against the critical temperature obtained from the simulation model. The critical temperature of the product gradually increases until the mass percentage is at 50% and then increases steadily as the highest critical temperature recorded is estimated at 331.5 °C. The increase in the critical temperature is also evidence that the strength of the product has increased, which can be seen as an advantage of the dissolution/reprecipitation method.

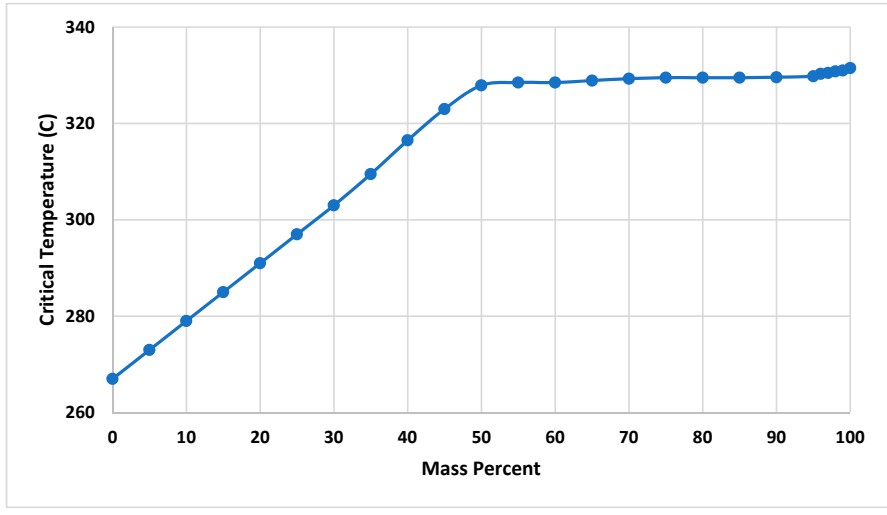

**Figure 4.** Plot of the critical temperature against the mass percentage of the product.

The true critical temperature of the product stream is recorded at 327.9 °C, and the true critical pressure is at 4940 kPa. The performance of the reprecipitation column has also been analysed using Aspen HYSYS. Figure 5 shows how the different components' composition changes through the trays of the column from top to bottom. The plot shows that the mole fraction of LDPE is its lowest value at the top of the column, while the PetE used as the non-solvent is its highest. At tray 10, the LDPE mole fraction increases as the stream flow down the column while the PetE mole fraction decreases, reaching 0.5 mol for both components. As the reprecipitation process continues, the LDPE mole fraction increases rapidly as the stream reaches tray 20; on the other hand, PetE composition decreases at the same pace. With the stream reaching the bottom of the column, the plot shows that the stream composition is 99% LDPE and less than 1% PetE. This is evidence of the great performance of such a recycling technique to obtain pure LDPE. The other components of the stream, such as Turpentine/PetE, are neglected since they are minimal traces in the stream.

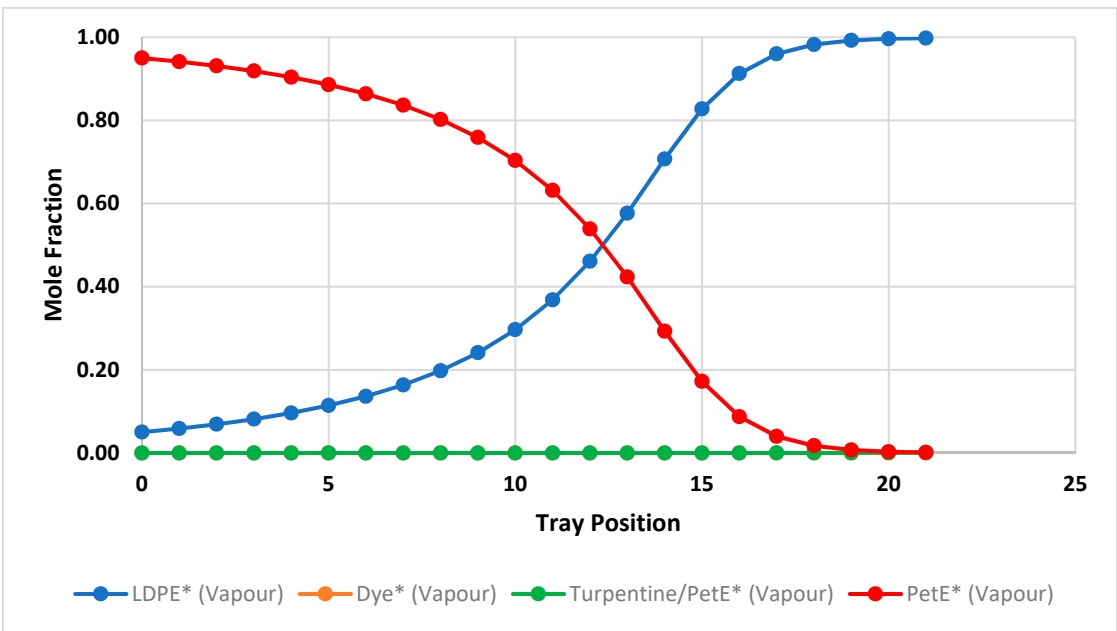

**Figure 5.** Composition against Tray position from the top of the reprecipitation column.

The simulation model developed showed results consistent with the literature [21]. Only a higher conversion rate is obtained when only waste LDPE is treated. The simulation model was developed and improved regularly to achieve the highest conversion rate and accuracy. Several simulation attempts took place with the use of different methods and simulation techniques. The attempts are recorded and shown in the appendices section of this report. Table 11 gives thorough details on this, which indicates the clear difference in the nomenclature of the new product as associated with density at the Condenser unit (6.918 kgmole/m$^3$) to that of the Reboiler unit (29.736 kgmole/m$^3$).

According to Table 11, the negligible traces of impurities can be derived from the differences exhibited in the mole weight, as clearly seen between the cases Reboiler (28.274) and the condensing column (93.886).

**Table 11.** Column properties vs. tray position from top for liquid.

| | Surface Ten [dyne/cm] | Mole Weight (Lt Liq) | Density (Lt Liq) [kgmole/m$^3$] | Viscosity (Lt Liq) [cP] | Therm Cond (Lt Liq) [W/m-K] | Heat Cap (Lt Liq) [kJ/kgmole-C] |
|---|---|---|---|---|---|---|
| Condenser | 14.133 | 93.886 | 6.918 | 0.225 | 0.102 | 224.503 |
| Tray 1 | 14.227 | 93.168 | 6.986 | 0.226 | 0.102 | 222.842 |
| Tray 2 | 14.341 | 92.307 | 7.069 | 0.227 | 0.102 | 220.852 |
| Tray 3 | 14.477 | 91.270 | 7.172 | 0.229 | 0.102 | 218.452 |
| Tray 4 | 14.642 | 90.009 | 7.301 | 0.230 | 0.102 | 215.536 |
| Tray 5 | 14.844 | 88.465 | 7.463 | 0.233 | 0.102 | 211.965 |
| Tray 6 | 15.093 | 86.555 | 7.673 | 0.236 | 0.102 | 207.546 |
| Tray 7 | 15.403 | 84.164 | 7.950 | 0.239 | 0.102 | 202.016 |
| Tray 8 | 15.792 | 81.135 | 8.326 | 0.244 | 0.102 | 195.012 |
| Tray 9 | 16.287 | 77.252 | 8.855 | 0.249 | 0.102 | 186.029 |
| Tray 10 | 16.919 | 72.229 | 9.631 | 0.257 | 0.102 | 174.404 |
| Tray 11 | 17.717 | 65.748 | 10.821 | 0.267 | 0.102 | 159.397 |
| Tray 12 | 18.681 | 57.662 | 12.712 | 0.279 | 0.102 | 140.659 |
| Tray 13 | 19.708 | 48.539 | 15.669 | 0.291 | 0.102 | 119.501 |
| Tray 14 | 20.564 | 40.119 | 19.693 | 0.297 | 0.101 | 99.971 |
| Tray 15 | 21.078 | 34.173 | 23.814 | 0.296 | 0.101 | 86.197 |
| Tray 16 | 21.312 | 30.865 | 26.832 | 0.291 | 0.101 | 78.549 |
| Tray 17 | 21.406 | 29.288 | 28.520 | 0.287 | 0.101 | 74.907 |
| Tray 18 | 21.443 | 28.601 | 29.318 | 0.285 | 0.101 | 73.319 |
| Tray 19 | 21.456 | 28.328 | 29.649 | 0.284 | 0.101 | 72.680 |
| Tray 20 | 21.460 | 28.247 | 29.755 | 0.284 | 0.101 | 72.480 |
| Reboiler | 21.457 | 28.274 | 29.736 | 0.284 | 0.101 | 72.515 |

### *3.1. Crystallinity and Melting Ranges*

The crystallinity and melting point of the virgin and waste LDPE before and after recycling were calculated. The melting and crystallinity point values for LDPE used in laboratory experiments are shown in Table 12. The recycling procedure did not affect the melting point and stayed within the permissible limits. The melting temperature and melting range fluctuation for the waste and recycled LDPE can be credited to plasticisation and the additives present in the polymer, which can be due to the presence of tiny traces of the solvent that will probably remain in the polymer structure [21,41]. The same results obtained from the literature are assumed to be applied to the simulation model.

**Table 12.** Melting temperature and crystallinity of virgin, waste and recycled LDPE studied [17].

| Polymer | Melting Temperature | | | Crystallinity% | | |
|---|---|---|---|---|---|---|
| | Virgin | Waste | Recycled | Virgin | Waste | Recycled |
| **LDPE** | 106 | 113 | 113 | 52 | 26 | 50 |

A restoring process is conducted, ranging between 24 and 40% for LDPE. Once the restoration procedure is finished, an increase in the crystallinity is observed from 32 to 44.1%. The variation noticed can be due to the solution under small cooking conditions, indicating that the recycling process serves as a type of annealing treatment [21].

### *3.2. Mechanical Properties*

The results of the tensile breaking tests performed in a laboratory environment showed that the tensile stress of the polymer at maximum yield and load increased after recycling

and became similar to that of the virgin polymer. In addition, the elasticity tests showed that it also became very similar to the original value after the recycling process. At the same time, the strain at break was lower than that of the waste LDPE [17]. This occurrence is attributed to the role of the additives that are present in the waste LDPE, in addition to the influence of the fractionation phenomena that happens during the dissolution and reprecipitation process. This means that some lower molecular weight fractions may remain soluble in the solvent/non-solvent phase [18]. These same mechanical properties investigated in a laboratory environment are assumed to be the same in the simulation developed.

*3.3. Cost*

The cost of utilities and equipment can be estimated using the Aspen Process Economic Analyser, supported in the authors' other work [25]. The cost of equipment such as heaters, coolers, tanks, and distillation columns are estimated to be £608,000 collectively, with an installation cost of £1,270,000. Coolers and heaters are added to the process to decrease the amount of energy needed by the distillation column to achieve the highest recovery possible. The presence of such utility equipment will decrease the workload that the distillation column will require. The accuracy of these estimations might vary in real life since a fair number of assumptions are in place to ensure a smooth simulation. The total cost of the process can still be decreased further by optimising the process. Thorough cost estimation will still be needed to accurately calculate the entire process's capital cost, including equipment cost, utility cost, installation cost and other variables that can affect the cost of the process. The solvent/non-solvent system cost must also be considered to accurately predict the net profit of such a process with the addition of knowing the prices of the products produced. In addition, oil prices can rise, leading to even more interest in investment and expansion. In the same way, the production of hydrogen and carbon nanotubes from natural gas has attracted research and industrial interest [42–48].

## 4. Conclusions

Aspen HYSYS was used to simulate the restoration of waste LDPE using the dissolution/reprecipitation technique, which has been proven to be feasible in many experiments but not on full-scale commercial plants. This is due to several challenges, one being that the process is not considered very economical. Generally, most waste plastics are mixed. Therefore, the main challenge that comes into thinking is the separation and recycling of components individually. The dissolution of plastics in solvents is considered a complex process. It differs from individual plastic due to the interactions between the solutes. The model developed serves as a reference system to predict product yield, product composition and process behaviour and response to operating factors, including changes in temperature effect, pressure, and flow rate. The selection of the ideal solvent/non-solvent is significant. The solvent/not-solvent system selected for the simulation showed immense potential in the laboratory environment and had the highest recovery rate. Therefore, these solvents were implemented in the simulation process. The fact that the solvent/non-solvent system is not hazardous gives it a considerable advantage since hazardous solvents have limited usage. Choosing low poisonous and much cheaper extractors such as terpene oils is the future direction for recycling waste plastic and related polymers.

A solvent (Turpentine/PetE) is added to the waste polymer to eliminate the additives present in LDPE, thereby setting the stage right for the increased molecular weight of the yield. The polymer is assumed to be precipitated as a powder or small grain by adding an antisolvent. The simulation is based on Hadi et al.'s [17] researched work and showed excellent polymer recoveries with the precipitated stream compositions consisting of 99% pure LDPE, which is higher than the composition of the recovered LDPE from experiments using the same solvent/non-solvent system and the same technique.

It is also assumed that the solvent/non-solvent is recycled into the process to reduce the cost of the process. Aspen HYSYS estimated the capital cost of the process by calculating

equipment, utility, and installation costs. Many other aspects are needed to be considered, such as the feedstock's availability. The simulation model developed shows that it is feasible to recycle huge amounts of waste plastics, more than 1000 tonnes per year, with higher recovery rates than the small-scale laboratory experiments. This implies that such a process can treat huge amounts of waste plastics that would have gone to disposal. More research is needed to improve the process of dissolution/reprecipitation of waste plastics. Dissolution and reprecipitation importance are one of the most recommended processes for recycling waste plastics considering the diversity of contaminants and materials in such wastes.

**Author Contributions:** Conceptualisation, Supervision, Writing—Original draft preparation, Reviewing and Editing, S.H.Z.; Investigation, Data curation, Writing—Report, A.A.H.; Writing revision, Reviewing and Editing, O.Y.Y.; Reviewing and Editing, A.A.J. All authors have read and agreed to the published version of the manuscript.

**Funding:** This research received no external funding.

**Data Availability Statement:** Not applicable.

**Conflicts of Interest:** The authors declare no conflict of interest.

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
