# Peer review of "Modelling and Simulation of Dissolution/Reprecipitation Technique for Low-Density Polyethene Using Solvent/Non-Solvent System"

_processes, doi:10.3390/pr10112387_

Round 1
Reviewer 1 Report
The authors performed the simulation of a recovery process of low-density polyethene. However, the manuscript needs to be substantially improved before publication addressing the following items.
- Introduction needs to be improved focusing on chemical recycling of plastic waste and also simulation modelling. It is important to review recent literature in the field. What is the novelty of this work?
- What are the scientific contributions of paper?
- Experimental does not make sense in the manuscript because those are not performed by authors instead, they sourced it from a published article. Same for some sub-sections and tables. The paper seems like a duplication of the published paper.
- Does the data in Table 3 sourced from literature? If yes, a reference should be provided.
- How authors justified the results obtained from the simulation?
- Specific temperature should be included in Figure 1 for each unit process.
- Line 369-370 “The physical properties of the precipitated LDPE, such as boiling point and melting point, can provide crucial information that can help identify the product to establish its purity.”
Does plastic (polymer) have a boiling point?
- What information Figure 2 has given? What do the abbreviations mean?
- Page 11: Why did molecular weight increase?
- How do authors conclude that the process is cost-effective without comparing it with other recycling methods?
- Conclusion should be revised focusing on the scientific contribution of this study.
Reviewer 2 Report
This is a highly interesting manuscripts, rather well done on an interesting and current topic, recommended to be published. Thera are however some minor comments to be made:
Line 27 – The sentence beginning with “ Plastics are ..” is unclear, with unfortunate use of terms. Perhaps to be changed to “Plastics are likely the most versatile material known to mankind”?
Line 43 – The terms polymers and plastics are used without appropriate distinction, which needs to be improved. First, plastics are typically understood to be polymeric materials consisting of polymers and additives. A polymer is not likely a plastic since additives are generally needed to make a useful material. Secondly, plastics are typically made of synthetic polymers, as polymeric materials based on natural polymers (such as starch), are unlikely called plastics. The sentence should be improved to make theese distinctions clear.
Line 47 – It says “Polymers are derived from petroleum ..”. This is not true in all cases. To a minor extent we do have other sources.
Line 69 – It says “ …will lead to an increase …”. We do not know this, but it would be acceptable to read that it will likely lead to that.
Line 103 – It is stated that Turpentine/petroleum ether was choosed for the study, but not explained why. On lines 257-260 it is stated that this combination had highest recovery, valid in a rather narrow perspective (Table 2). Can it be further explained why Turpentine/petroleum ether was chosen?
Line 447 – Is the sentence incomplete or is the capital letter starting on line 451 wrong?
Round 2
Reviewer 1 Report
I accept the corrections made.